

# Four years of atmospheric mercury records in Northwestern Patagonia (Argentina): potential sources, concentration patterns and influence of environmental variables observed at the GMOS EMMA station

María C. Diéguez[1], Patricia E. Garcia[1], Mariantonia Bencardino[2], Francesco D'Amore[2], Jessica Castagna[2], Sergio Ribeiro Guevara[3] and Francesca Sprovieri[2]

[1] Laboratorio de Fotobiología, INIBIOMA, CONICET-Universidad Nacional del Comahue, Quintral 1250, 8400, Bariloche, Argentina

[2] CNR-Institute of Atmospheric Pollution Research, Rende, Italy

[3] Laboratorio de Análisis por Activación Neutrónica, Centro Atómico Bariloche, Comisión Nacional de Energía Atómica, Avda. Bustillo km 9, 8400, Bariloche, Argentina

Correspondence to: María C. Diéguez (dieguezmc@gmail.com)

## Abstract

The Global Mercury Observation System (GMOS) project, has developed a global-scale network of ground-based atmospheric monitoring sites, expanding the coverage of atmospheric mercury (Hg) measurements worldwide and improving the understanding of global atmospheric Hg transport and deposition, particularly in regions of the South Hemisphere where atmospheric Hg observational data is limited. This work provides the first continuous records of gaseous elemental Hg (GEM) concentrations observed from October 2012 to May 2016 in Northwestern Patagonia (Argentina) at the GMOS EMMA monitoring station (41°7'43.82"S, 71°25'11.89"W, 803 m a.s.l). The monitoring site is located inside Nahuel Huapi National Park, a natural reserve in the Lake District of Andean Patagonia. The area is within the Southern Volcanic Zone, influenced by several active volcanoes aligned in the Andes cordillera. During the studied period, GEM concentrations ranged between 0.23 and 1.43 n gm$^{-3}$, with an annual mean of $0.9 \pm 0.15$ ng m$^{-3}$. GEM records at EMMA station resemble background concentrations measured in Antarctica and other remote locations of the Southern Hemisphere. GEM concentrations showed seasonal variation with mean values higher during spring ($0.93 \pm 0.13$ ng m$^{-3}$) and winter ($0.92 \pm 0.10$ ng m$^{-3}$) followed by summer ($0.86 \pm 0.15$ ng m$^{-3}$) and at last by autumn ($0.81 \pm 0.15$ ng m$^{-3}$). Further, a clear daily pattern



was observed, with higher GEM levels during day-time than at night-time across all seasons. Multivariate analyses showed that GEM levels are chiefly determined by meteorological parameters, and, in particular by the westerly winds which represented the most influential variable on GEM records. In order to investigate the potential impact of natural and/or anthropogenic emission sources as well as the role played by the long-range transport on GEM levels, analyses of HYSPLIT

backward trajectories (BWT) were carried out for different periods characterized by low and high GEM concentrations. The BWT analysis highlighted the influence of clean oceanic air masses and also of the local and regional active volcanoes in the Andes cordillera.

**Keywords**: Gaseous elemental mercury, Atmospheric transport, GMOS, North Patagonia

## 1 Introduction

Mercury (Hg) is a pollutant with long-range atmospheric transport that determines its enormous potential to reach out and deposit in regions far from Hg sources (Pirrone and Mason, 2009; Pirrone et al., 2010; 2013; Driscoll et al. 2013). The biogeochemical cycle of Hg is complex involving its transformation and transfer in the atmosphere, lithosphere and

hydrosphere under the influence of several environmental variables (reviewed in Selin, 2009; Driscoll et al., 2013; Krabbenhoft and Sunderland, 2013). Gaseous elemental Hg (GEM or $Hg^0$) can be transported after emission for long distances whereas gaseous oxidized Hg (GOM) and particle bound Hg (PBM) have a shorter residence time in the atmosphere and thus, are readily deposited to terrestrial and aquatic systems (Driscoll et al., 2013). In soils and water Hg can be present in inorganic forms and also can convert into methylated species (MeHg and DMeHg), neurotoxins that bio-

accumulate in aquatic food webs, ultimately affecting humans and wildlife (Ullrich et al. 2001; Horvat, 2002). Currently, Hg contamination is recognized as a global t by the countries subscribing the Minamata Convenion, a global treaty to reduce Hg pollution (http://www.mercuryconvention.org/).

Several investigations in different locations of South America (SA) support the idea that Hg may pose a problem at a

regional scale (UNEP, 2009; 2013). However, global Hg models have large uncertainties about Hg levels in this region since estimates are mostly extrapolated from regional grouping not always reflecting local situations because measurements or statistical information are not available (AMAP/UNEP, 2015; Evers et al., 2016; Kwon and Selin, 2016). In general, in some countries of SA Hg pollution has been attributed mostly to sustained artisanal gold mining (Cooke et al., 2009). However, in pristine remote areas of the region the occurrence of moderate to high Hg levels has been related with

eventual inputs from geogenic sources, biomass burn and atmospheric transport and deposition (Nriagu and Becker, 2003; Ribeiro Guevara et al., 2010; Daga et al., 2014; 2016a,b; Hermanns and Biester, 2013a,b; Higueras et al., 2014).





In Patagonia, the southernmost area of SA comprised in the Southern Volcanic Zone (Naranjo and Sterns, 2004), there is increasing evidence arising from lake sediment studies reporting high Hg levels corresponding with volcanic events in the region, and also sustained Hg contributions through atmospheric transport and deposition (Ribeiro et al., 2010; Hermanns and Biester, 2013a,b; Daga et al., 2014; 2016a,b).

In particular in Nahuel Huapi National Park (NHNP, Northwestern Patagonia, Argentina), a well-protected natural reserve, several investigations have highlighted the occurrence of Hg in air and freshwater bio-indicators (Ribeiro Guevara et al., 2004a; 2004b), and moderate to high Hg levels in lake biota (Arribére et al., 2010; Rizzo et al., 2011; 2014; Arcagni et al., 2016). The presence of high Hg levels in different ecosystem compartments has been attributed alternatively to eventual inputs from several active volcanoes aligned in the Andean stretch and to biomass burn (wild fires) (Ribeiro et al., 2010;

Daga et al., 2014; 2016a,b). However, the fact that most total dissolved Hg in freshwater ecosystems of the area is in the inorganic form (97%) (Rizzo et al., 2014), suggests that atmospherically transported Hg and deposition in catchments sustain baseline Hg levels, coinciding with evidence from lake sediment records (Daga et al., 2016a,b and references therein).

In contrast with the increasing evidence of Hg occurrence within ecosystem compartments of the Patagonian region, the identification and measurement of the potential sources of Hg, such as precipitation and atmospheric transport, is at its

starting point. Until recently, atmospheric Hg levels in the region were derived from point measurements performed in Northwestern Patagonia, reporting a mean GEM concentration of ~7.5 ± 1.4 ng m$^{-3}$ (Higueras et al., 2014).

In 2012, the GMOS EMMA station was established inside Nahuel Huapi National Park, as part of the Global Mercury Observation System (www.gmos.eu), which has expanded the coverage of atmospheric Hg measurements by establishing Hg

monitoring stations around the globe and in strategic remote locations of the SH, including different sites in SA (Sprovieri et al., 2016a,b). New stations such as the EMMA in Patagonia, are providing insight into the concentrations and patterns of Hg levels in remote regions not previously covered in other networks. In particular, the EMMA station is one of the five GMOS sites spotted in Central and South America and the southernmost Hg monitoring site of the continent, providing novel regional information of Hg levels useful for Hg global models (Sprovieri et al., 2016a, Travnikov et al., 2016). The present

work provides the first continuous records of GEM concentrations measured from October 2012 to May 2016 at the GMOS EMMA station in Northwestern Patagonia. High frequency GEM concentration data recorded under the standard operational procedures of the GMOS project has been analyzed to provide a description and to discuss: i- GEM concentrations and seasonal and daily patterns observed during the sampling period; ii- the influence of local environmental parameters on GEM behavior, and, iii-the potential sources of the atmospheric Hg transported by air masses using backward trajectory

analysis.

## 2 Experimental

### 2.1 Site description



The EMMA Station is located at 41°7' 43.33"S, 71°25'12.03"O; 800 m a.s.l, in the central area of the natural reserve Nahuel Huapi National Park (NHNP; 40.145-41.592°S; 71.028-71.966°W) within the lake district of Northwestern Patagonia (Argentina) (Fig. 1). The station was established as a GMOS site in 2011 in the field research station of the Photobiology Laboratory (INIBIOMA-CONICET), starting its operation in October 2012.

The whole stretch corresponding to North Patagonia is included in Southern Volcanic Zone of the Andean Volcanic Belt (Naranjo and Sterns, 2004), including several active volcanoes aligned in the latitudinal range from 33°S to 46°S. NHNP limits at its western side with the active volcanic formation Puyehue Cordón Caulle. In recent years, major eruptions of the volcanoes Chaitén (2008-2010), Puyehue Cordón Caulle (2011) and Calbuco (2015) impacted the region (Ribeiro Guevara et al., 2010; Daga et al., 2010; 2014; 2016a,b).

The climate of North Patagonia is strongly modulated by the presence of the Andes cordillera and has been characterized as cold-temperate with well-defined wet and dry seasons (Paruelo et al., 1998; Rusticucci et al., 2014; Bianchi et al., 2016). The South Pacific westerly air current carries humid winds from the Pacific Ocean to the continent which discharge their humidity as they pass over the Andes, becoming dry in few kilometers and limiting the Atlantic influence (Garreaud, 2013). Overall, these conditions result in a pronounced climatic contrast between the Pacific side of the Andes and the Atlantic side,

and also in areas within the east Andean stretch, characterized as an extremely sharp altitudinal and climatic gradient, affecting particularly precipitation. These extremes are represented in the area of Nahuel Huapi National Park within a longitudinal distance of about 70 km from the cordillera, with and altitudinal decline from ~3000 to 670 m a.s.l and precipitation grading from 3500 mm y$^{-1}$ close to the Andes to 800 mm y$^{-1}$ in the sierra and meseta stretches. Such environmental conditions echo on vegetation which grades from the deciduous and evergreen *Nothofagus* forests typical of

Andean valleys to a shrub-graminous steppe at the lower side in the sierra and meseta areas (Mermoz et al., 2009). The Park includes the headwaters of the largest freshwater network of North Patagonia, encompassing high mountain lakes and streams and deep and shallow lakes at the piedmont, most belonging to the Nahuel Huapi catchment (Fig. 1).

There are three population settlements in the Park on the shores of Lake Nahuel Huapi (557 km$^2$), San Carlos de Bariloche city (population ca. 115,000), and the villages of Dina Huapi (ca. 4,000 inhabitants) and Villa La Angostura (ca. 15,000

inhabitants). The main economic activity within the Park is tourism which concentrates mostly in winter and summer.

**2.2 Data acquisition and handling**

The meteorological parameters, atmospheric pressure, air temperature, wind speed and direction, dew point, precipitation, and relative humidity were continuously recorded at the EMMA station by an integrated weather station (Davis Vantage Pro)

located besides the Hg monitoring instruments. Ancillary meteorological data was uploaded periodically in the GMOS website (G-SDI) (http://www.gmos.eu/sdi).

GEM concentrations have been continuously carried out using an automated Hg vapor analyzer Tekran 2537B (Tekran Instrument Corp., Ontario, Canada) located inside a shelter (room temperature 15° ± 4 °C) which pumps ambient air from the outside through a Teflon sampling line. The system applies a specific set of collection by amalgamation on dual gold



cartridges used alternately over 5 minutes and thermal desorption (500°C) procedures, resulting in operationally defined continuous releasing GEM from the traps into a carrier gas (ultra high purity argon) and its quantification by Cold Vapor Atomic Fluorescence Spectroscopy (CVAFS). The detection limit is 0.1 ng m$^{-3}$ at a flow rate of 1 L min$^{-1}$. The instrument performs automatic internal permeation source calibrations every 71 hours, and is regularly assisted to secure its

performance following the standard operational procedures (SOPs ) of the GMOS project (Sprovieri et al., 2016). Routinely, GEM data collected in the EMMA station is logged automatically into a computer using the software Tekcap® (Tekran Instrument Corp.) and uploaded manually in the GMOS website. The raw GEM data obtained are then sorted out following the quality assurance and control (QA/QC) procedures in the GMOS data validation application G-DQM (Cinnirella et al., 2014; D´Amore et al., 2015). All of the in situ measured parameters presented in this work are reported at local time

(Bariloche Standard Time: UTC - 4.00h), and all concentrations refer to STP conditions.

### 2.3 Data processing and Statistical analysis

GEM validated data through the QA/QC procedure in the G-DQM application (GMOS data portal: http://www.gmos.eu/sdi/) was used for all the calculations and statistical analyses performed subsequently. The diel pattern of Hg was assessed

seasonally, considering the photoperiod (day-light hours) of the mid-season (summer: 07:00-20:30 h; fall: 9:00-18:30 h; winter: 09:00-18:30 h; spring: 07:00-20:30 h) to calculate day and night GEM concentrations between October 2012 and May 2016. The hourly-averaged GEM concentration was also computed for the study period.

A multivariate exploratory study including a cluster analysis (CA; Ward´s method and correlation as distance measure) and a principal component analysis (PCA; correlation matrices) was performed to determine the most influential meteorological

variables relating with GEM concentration, using R  framework (version 3.2.2) with packages pvclust (version 2.0-0) for CA and FactoMineR (version 1.31.4) for PCA.

Two way analysis of variance (2-Way ANOVA) was performed to study the day-night differences in GEM concentrations among seasons, considering again the hourly-averaged value for the complete time series (October 2012-May 2016) and using the software SigmaStat. Pos-hoc tests (Holm-Sidak) were performed to study day-time and night-time differences in

GEM concentrations patterns among seasons.

In order to determine the path of air-masses reaching the EMMA station, the hybrid single-particle Lagrangian integrated trajectory model (HYSPLIT) available at the NOAA Air Resources Laboratory (Air Resources Laboratory 2010) was used to calculate GEM backward trajectories (BWT) (Draxler and Rolph, 2003). Specifically, the BWT were calculated in correspondence with the highest and the lowest GEM concentrations events, using the READY Website

(http://www.arl.noaa.gov/ready.html). Calculations were performed for a total run time of 48 h, setting the start of a new BWT every 2 h, for a total number of 24 trajectories. The Global Data Assimilation (GDAS) set was used as meteorological input. The trajectory arrival height was established at the elevation of the EMMA station, 800 m above ground level, approximately the boundary layer height where pollutants are usually well mixed, thus allowing to discriminate the influence of atmospheric transport from local and regional sources. A label interval of 6 hours was also set. (see Fig. 6).



## 3 Results and discussion

### 3.1 GEM concentration levels and seasonal pattern analysis

Figure 2 shows GEM levels from October 2012 to May 2016 with concentration ranging between 0.23 and 1.42 ng m$^{-3}$ with
an average of 0.87 ± 0.15 ng m$^{-3}$. Table S1 reports a statistical summary of GEM concentration for both, the whole sample
period as well as for each of the four seasons, calculated from validated GEM observations. The general lack or limited
availability of previous observational data of atmospheric Hg for SA and, in particular for Patagonia, hinders a direct
comparison of the data obtained during the studied period. In spite of this, however, atmospheric Hg measurements performed
at five GMOS sites located in the SH made a first comparison possible. The resulting mean GEM concentration recorded at
the EMMA station is within the concentration boundaries reported for the SH (0.84 -1.03 ng m$^{-3}$) (Sprovieri et al. 2010a,b;
2016a; Slemr et al., 2015; AMAP/UNEP, 2015; Angot et al., 2016). However, they are closer to the mean values recorded at
the Antarctic GMOS sites Dumont D´Urville and Concordia Station (Sprovieri et al., 2016a). In a more general context, the
median GEM concentration recorded in Northwestern Patagonia during the studied period (0.87 ng m$^{-3}$, ranging seasonally
between 0.80 and 0.93 ng m$^{-3}$) falls in the lower extreme of the Northern-Southern Hemispheric gradient computed from
records of 2013 and 2014 at GMOS sites. The hemispheric gradient encompasses median GEM concentrations ~1.5 ng m$^{-3}$ in
the North Hemisphere, ~1.2 ng m$^{-3}$ in the Tropics and ~0.93-0.97 ng m$^{-3}$ in the South Hemisphere (Sprovieri et al., 2016a), in
line with previously described hemispheric trends (Lindberg et al., 2007; Sommar et al., 2010; Sorensen et al., 2010 a,b;
Sprovieri et al., 2010a).

The mean GEM concentrations were higher in spring and winter, followed by summer and autumn levels (Table S1; Fig.S2).
Concentration peaks in the hourly averaged GEM were recorded at different times of the day; ~around 11:00 in summer,
between 12:00 and 15:00 in autumn, from 11:00 to 13:00 in winter and between 09:00 and 12:00 in spring (Fig. S2). Hourly
averaged GEM concentrations were significantly higher during day-time (day-time and night-time hours defined by the local
photoperiod at mid-season) than at night (ANOVA, F=532.82; p<0.001). This pattern was similar across seasons although
the trend was more pronounced in both summer and autumn (F=1017.83; p<0.001; Fig. 3). Remarkably, mean day-time
GEM concentrations in spring and winter were similarly higher (p>0.05) as compared to the lower mean levels found in
summer and even lower in autumn (p<0.05). Night-time GEM concentrations were higher in spring followed by winter,
decreasing towards summer and autumn when the lowest mean concentration was recorded (Fig. 3).

### 3.2 Meteorological data behavior at the EMMA station

During the study period temperature fluctuated between -9.6°C and 37.7°C, averaging 9.6°C ± 7.8°C. A marked thermal
seasonality was observed which reflected in the contrasting mean summer and winter temperatures (14.5± 7.9°C and 2.2 ±
4.4°C, respectively) (Table S1). The relative humidity showed high values during autumn and winter (~71%) compared to
spring and summer (~58%) (Table S1).





Wind speed (ws) showed higher values during summer and spring ($4.3 \pm 4.09$ m s$^{-1}$ and $4.1 \pm 4.5$ m s$^{-1}$, respectively) (Table S1, Fig. S2). The predominant winds bring air masses from the Pacific as indicated by the high frequencies showed by the wind records from NW (315°; frequency=15387) and WNW (292.5°; frequency=13503) (Fig. S2). In spring, autumn and winter the predominant winds were from the NW, while in summer alternated between WNW (n = 5080) and NW (n=4308)

(Fig. S2a-e). The NW winds were more frequent in spring and summer compared to autumn and winter, increasing during daytime from around 8:00 h up to 16:00 h to 19:00 h depending on the season and decreasing afterwards (Fig. S2f). In contrast, WNW winds were more frequent in summer and autumn, showing higher values around 12:00 and 20:00 h, whereas the influence of this wind direction was similarly low in winter and spring (Fig. S2g). The recorded wind pattern at the EMMA station resembles to generalized trends described for Andean Patagonia in the literature (Paruelo et al., 1998;

Garreaud et al., 2013; Rusticucci et al., 2014).

**3.3 Relationship between meteorological variables and GEM concentrations**

The multivariate exploratory analyses performed to study the relationship between the meteorological variables and GEM, pointed out the strong influence of wind direction and, secondarily, of wind speed and temperature on GEM concentration.

The cluster analysis (CA) showed four groups of variables, the first including relative humidity, the second with wind direction and GEM, the third grouping wind speed and temperature, and the last group gathering the second and the third ones (Fig.4a). Therefore, it is apparent that wind direction is a driver of GEM concentration, while temperature and wind speed are also influential meteorological variables. The principal component analysis (PCA) performed including the same variables reinforced the pattern revealed by the CA. Two factors accounted to explained ~68% of the variance, and the

inclusion of a third factor increased the resolution of the model explaining ~85% of the variability during the studied period (Fig. 4b; Table S2). Factor 1, explaining ~45% of the variance, denotes the importance of the dispersion process as indicated by the direct contribution of the wind speed and, at a much lesser extent, of the wind direction to the component (Table S2). However, local meteorological conditions such as relative humidity and temperature are also relevant. Factor 2 explained ~23% of the variability with the contribution of the variables wind direction, wind speed and temperature.

Overall, the results of the multivariate exploration indicated the influence of the dispersion processes in controlling mercury levels recorded at the EMMA Station.

To better understand the influence of wind, we correlated the wind frequency of the prevailing directions, NW and WNW, and GEM concentrations across the whole study period (October 2012-May 2016) and seasonaly. The results showed no significant correlation between the frequency of NW and WNW winds and GEM for the complete dataset (p>0.05) (Fig. S3).

However, a direct correlation was found between the frequency of NW and WNW winds and GEM concentration in the summer (r=0.45, p=0.027 and r=0.46, p=0.02, respectively) and autumn (r=0.64, p=0.0007 and r=0.63, p=0.001, respectively).



Remarkably, considering the whole dataset, the GEM concentrations were found to be directly and significantly correlated with the NW wind speed. However, the correlation factor was too low to allow describing a pattern (r=0.13, p<0.001). No clear relationship was found between WNW wind speed and GEM concentration (p>0.05). When the same analysis was performed seasonally, the correlation between NW wind speed and GEM was slightly stronger, particularly in spring

(r=0.37, p<0.0001) and winter (r=0.28, p<0.0001); whereas the correlation between WNW wind speed and GEM was more apparent in autumn (r=0.4, p<0.001) than in any other season (0.18 < r < 0.23; p<0.001).

Figure 5 summarizes the seasonal trend of GEM concentrations recorded at the EMMA station in Patagonia, highlighting: i-the marked influence of westerly winds, ii-the daily concentration pattern, with higher GEM levels during day-time than at night, and, iii-the lower GEM levels in autumn and a least contrasting day-night pattern.

Finally, the direct relationship between GEM concentration and temperature computed for the whole data set resulted weak, although it was positive and significant (r=0.132, p<0.001).

### 3.4 Air mass back-trajectory analysis and potential influence of regional sources

The backward trajectory (BWT) analysis showed that, both high and low GEM periods occurred under the influence of air

masses from westerly directions (Fig. 6). At low GEM levels, the BWT analysis pointed out the major influence of clean oceanic air masses coming from the west (W) and southwest (SW) (Fig. 6a-d). Low mercury concentration events were recorded during summer (February 2015 and 2016) and once in early autumn (April 2015). As highlighted in Figure 6a-d, these events were primarily related to air masses coming from the free troposphere (BWT starting from 2000-8000 m a.s.l) and resulting from a long range transport.

Conversely, high Hg level events were recorded during spring (2012 and 2013) and summer (2013 and 2014), occurred concomitantly with air masses coming from not a common direction and characterized by a lower elevation of origin and shorter range transport, thus involving a local/regional influence (see Fig. 6e-f). For all these trajectories a terrestrial influence can be further inferred, likely due to the effective barrier of the Andes cordillera that confront the westerly air masses changing their thermal and humidity properties as they cross the landmass (see Garreaud et al., 2013). This terrestrial

contribution may likely include and reflect the influence of the several active volcanoes aligned in the cordillera which are well known as sources of particulate and gaseous materials carried by the westerly winds that deposit in ecosystems at the eastern side of the Andes (Ribeiro Guevara et al., 2010; Bubach et al., 2012; Daga et al., 2014; 2016a,b; Higueras et al., 2014). At the particular geographic location of the EMMA station, the westerly winds arrive after discharging their humidity in the immediate west and east stretches surrounding the high altitudes, likely precipitating Hg in the stretch of the mountain

range and piedmont coinciding with the westernmost limit of Nahuel Huapi National, at ca.55 km at the west of the EMMA Station. Across this short longitudinal distance the precipitation grades from 3500 mm y⁻¹ close to the Andes to ca. 1000-1200 mm y⁻¹ at the EMMA Station (Mermóz et al., 2009; Bianchi et al., 2016). It is worth mentioning that higher Hg levels recorded at the EMMA Station occurred during the austral dry season comprising the last part of the spring and the summer which may be related with the fact that the lower precipitation volumes in the area could prevent Hg precipitation in the





boundaries of the cordillera and, thus, drier air masses could carry higher Hg levels to eastern locations of the Andean stretch.

## 4 Conclusions

This work provides the first long-term records of gaseous elemental mercury (GEM) concentrations performed in Argentina at the GMOS EMMA monitoring station. The obtained results contribute to Hg knowledge in describing the current Hg levels in Nahuel Huapi National Park (Northwestern Patagonia), a remote area of South America with past and present records of Hg impact from natural sources. GEM showed concentrations ranging from 0.23 and 1.43 ng m$^{-3}$ with an average of $0.865 \pm 0.149$ ng m$^{-3}$. These levels are in the lowest range within the concentration boundaries reported in the Northern-

Southern hemispheric gradient resulting from GEM monitoring at the ground-based sites of the GMOS network. The local GEM concentration pattern showed significant seasonal and daily trends, characterized by higher GEM levels in spring and winter and a daily concentration pattern displaying higher GEM day-time concentrations and lower night-time levels. Westerly winds showed up as the most influential variable directing GEM concentration and potentially contributing Hg. The BWT analysis, indicated that the influence of clean oceanic air masses coming from the west (W) and southwest (SW)

likely determine low GEM periods. In contrast, high GEM periods are apparently driven by air masses from western and southern directions which likely receive inputs of Hg while blowing through the Andes, supplied by the numerous active volcanoes aligned in the cordillera. The results presented here portray to date the status of the atmospheric Hg fluxes in the Northwestern Patagonia. Currently, Hg monitoring networks cover several regions of the NH, however the coverage in the SH is by far lower. The new GMOS ground-based monitoring sites in different regions, including several stations in the SH,

are expected to provide more comprehensive boundaries of Hg occurrence, and a better understanding of the environmental drivers interacting globally and regionally to help constraining the global biogeochemical cycle of this global pollutant.

## 5 Acknowledgements

This work was supported by the European Commission through funding of the 7[th] Frame Program to the Global Mercury

Observation System (GMOS) project (Grant Contract #26511); by International Cooperation Projects between CONICET (Argentina) and CNR (Italy) to MCD and FS; by CONICET PIP11220100100064 and Agencia FONyT PICT 2012-1200 and 2015-3496. Back trajectory analysis was possible through the kind support of the NOAA Air Resources Laboratory (ARL) through permission to use the facilities of the READY website (http://www.ready.noaa.gov).We are especially grateful to Dr. Mariana Reissig, and also to Julio Piacentini and Adrián Inchaurza.

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





**Figure captions**

**Figure 1.** Geographical location of the Global Mercury Observation System monitoring station EMMA (red point) in Nahuel Huapi National Park (area in pink) in Northwestern Patagonia (area in pink), Argentina.

**Figure 2.** Gaseous elemental mercury (GEM) concentration (ng m-3) recorded at the EMMA Station (Nahuel Huapi

National Park, Northwestern Patagonia) between October 2012 and May 2016. Reported values are GEM concentrations validated through the GMOS QA/QC process of the GMOS G-DQM application. Data gaps during periods of 2013 and 2015 are due to instrument failure and/or invalidated data. The dotted line depicts the mean GEM concentration of the period studied.

**Figure 3.** Seasonal trends in the mean GEM concentration (ng m-3) during day-time and night-time calculated from

continuous records from October 2012 to May 2016 at the GMOS EMMA Station (Nahuel Huapi National Park, Northwestern Patagonia). Letters on top of data columns indicate groups and differences among groups.

**Figure 4.** Multivariate analyses performed to study the relationship between meteorological parameters [temperature, relative humidity, wind direction and wind speed and the concentration of gaseous elemental mercury (GEM, ng m$^{-3}$) recorded between October 2012 and May 2016 at the EMMA Station (at the EMMA Station (Nahuel Huapi National Park;

Northwestern Patagonia). : a) Cluster Analysis plot, and, b) Principal Component Analysis (PCA) plot (Factor 1 vs. Factor 2).

**Figure 5.** Wind rose plots of seasonal hourly trends of gaseous elemental mercury (GEM) concentration recorded at the EMMA Station (Nahuel Huapi National Park, Northwestern Patagonia).

**Figure 6.** Backward trajectories (BWT) modelled (HYSPLIT, NOAA Air Resources Laboratory) for Low and High GEM

concentration levels recorded at the EMMA Station (Nahuel Huapi National Park, Northwestern Patagonia). BWT for low GEM concentration fluxes observed during: a) February 14, 2015; b) February 15, 2015; c) April 15, 2015 and d) February 19, 2016. BWT for high GEM concentration fluxes recorded during: e) November 14, 2012; f) January 14, 2013; g) November 7, 2013; h) January 18, 2014.





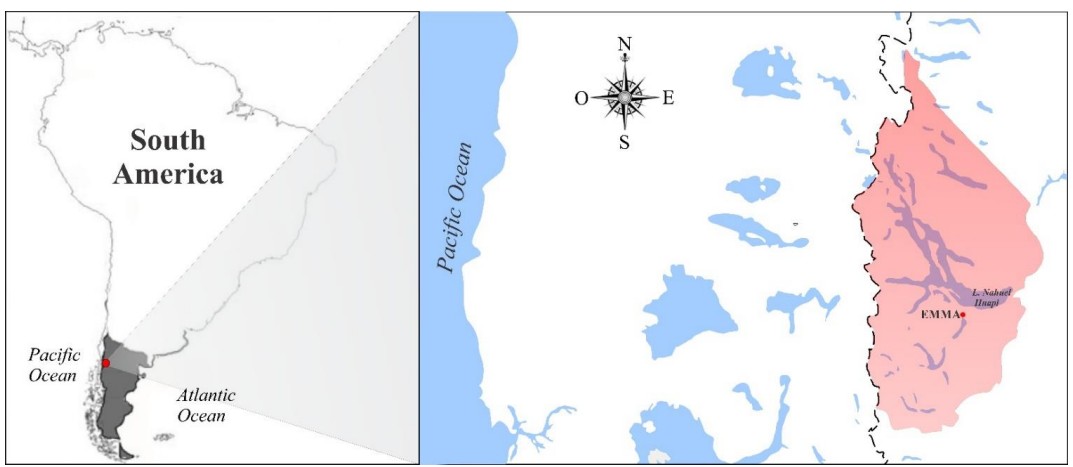

**Figure 1**

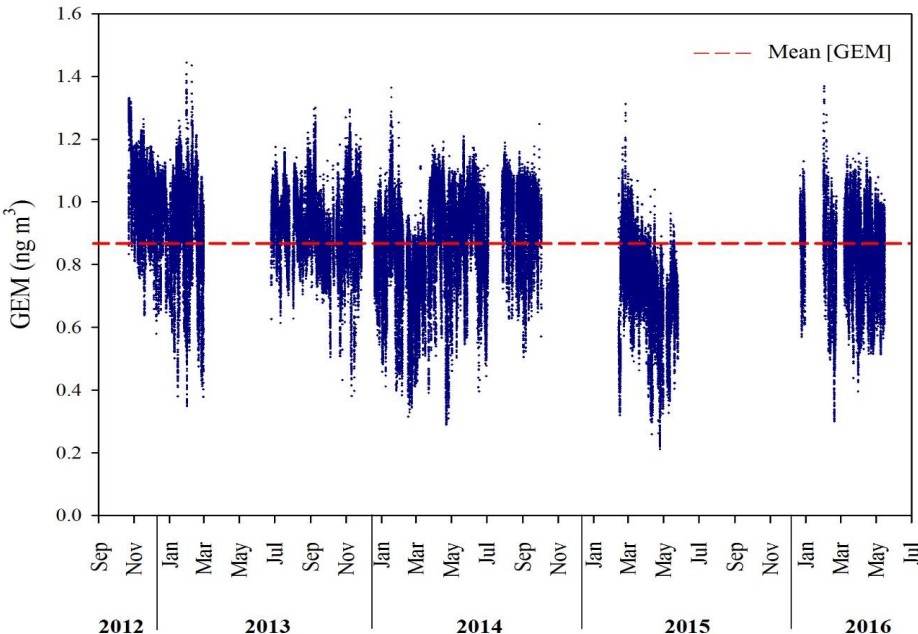

**Figure 2**





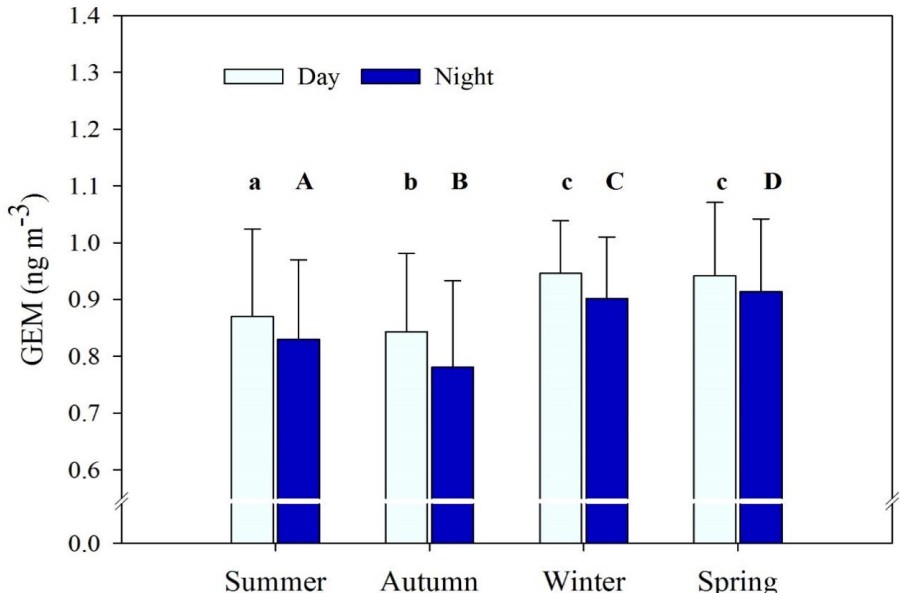

**Figure 3**

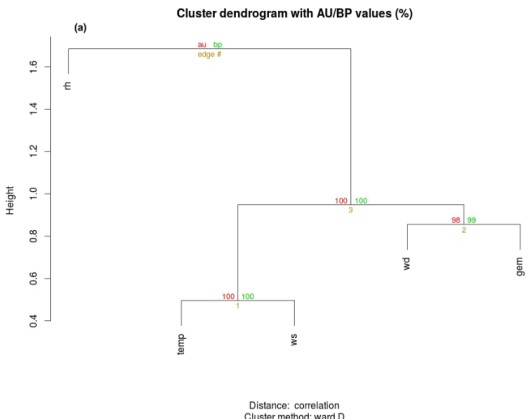

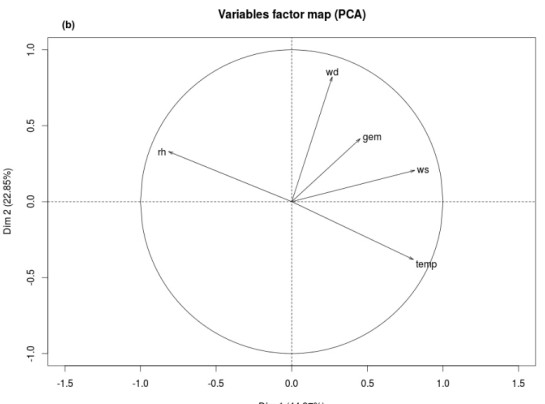

**Figure 4**





**Figure 5**





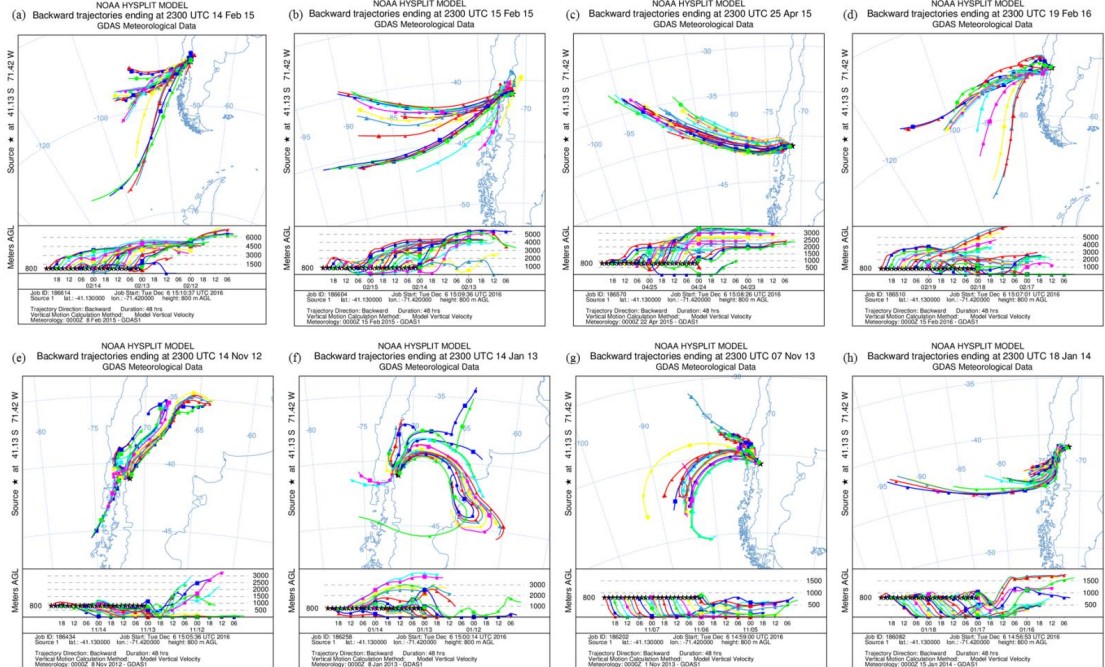

**Figure 6**