# Peer review of "Four years of atmospheric mercury records in Northwestern Patagonia (Argentina): potential sources, concentration patterns and influence of environmental variables observed at the GMOS EMMA station"

_Atmospheric Chemistry and Physics, 2016_

## Referee Comment (RC1) · Anonymous Referee #1 · 20 Feb 2017

Review of manuscript submitted for ACP, doi:10.5194/acp-2016-1076, 2017

Manuscript: Four years of atmospheric mercury records in Northwestern Patagonia (Argentina): potential sources, concentration patterns and influence of environmental variables observed at the GMOS EMMA station. From: María C. Diéguez , Patricia E. Garcia , Mariantonia Bencardino , Francesco D'Amore, Jessica Castagna , Sergio Ribeiro Guevara and Francesca Sprovieri.

[Figure]

This manuscript deals with a relevant issue, with proper instrumentation and linked to good analyrical protocols from the GMOS project. The manuscript reports gaseous elemental Hg (GEM) concentrations observed from October 2012 to May 2016 in Northwestern Patagonia (Argentina) at the GMOS EMMA monitoring station. It is a pity that the other Hg components of the mercury cycle (GOM and PBM) were not measured. I think the introduction needs to discuss that in more detail.

Also, in this study, only GEM and met variables were measured. The experimental design should have done to include other variables that could help to identify processes that influences GEM concentrations, other than only wind direction and air mass trajectory. Trace gases and simple aerosol measurements taking together with Hg are valuable enhancements. With only GEM and Met variables, it is hard to do a good scientific job. The cluster and PCA analysis had too small number of variables to deal with (only 4-5 variables), do not allowing a clear separation of the various components that influences GEM concentrations. Do you have precipitation amounts at the station? Or upwind of the station? This could help in the association between GEM and met variables.

With the study limitation, the conclusions are quite obvious: low GEM concentrations are observed when air masses comes from the ocean. High GEM concentrations come from volcanoes in the cordillera. This is the conclusion of the study, and is well known already from previous studies. What this study innovates in science? What is exactly the new issue brought to the knowledgebase? Only GEM concentrations of $0.865 \pm 0.149$ ng m$^{-3}$? No new Hg removal or production processes? No parallel measurements with other variables that could trace processes?

I think the manuscript needs a deep revision to make it to the published at the level of the others ACP papers.

Specific issues: Page 3 line 10: Instead of 'biomass burn", please use "biomass burning".

[Figure]

Page 3 Line 15: You report that "Until recently, atmospheric Hg levels in the region were derived from point measurements performed in Northwestern Patagonia, reporting a mean GEM concentration of ïA¿7.5 ± 1.4 ng m-3 (Higueras et al., 2014)". I think you must comment on this very high level reported. Why? Instrumental problems? Specific issues?.

Page 5 line 10: Your statement on STP is that "all concentrations refer to STP conditions.". Which STP conditions? There are several. Temperature of 0 or 25 degrees? Pressure? 1000 mb? Please specify.

Page 5 line 26 – for Backward Trajectories (BWT), the manuscript uses HYSPLIT plus Global Data Assimilation (GDAS). This could be the best possible approach, but the lack of meteorological stations in this part of the globe increases significantly the uncertainties of the modeled backward trajectories. I think you must recognize that and comment on the possible implications for the study. As you can see in Figure 6, air mass trajectories comes from Southern Pacific, with no met stations to validate the modeled air mass trajectories.

Page 8 line 10: You stated that: "Finally, the direct relationship between GEM concentration and temperature computed for the whole data set resulted weak, although it was positive and significant (r=0.132, p<0.001)". There is no "Weak" or "Strong" correlation, but instead, statistically significant or not. In this case, the GEM and temperature are statistically significant within 95% confidence interval. The question is why? Which mechanism could make sense for this association?

---

## Referee Comment (RC2) · Anonymous Referee #2 · 27 Feb 2017

Four years of atmospheric mercury records in Northwestern Patagonia (Argentina): potential sources, concentration patterns and influence of environmental variables observed at the GMOS EMMA station

The manuscript by Diéguez et al, Four years of atmospheric mercury records in Northwestern Patagonia (Argentina): potential sources, concentration patterns and influence of environmental variables observed at the GMOS EMMA station, reports on valuable measurements of atmospheric mercury in the southern hemisphere. As a global en-

vironmental concern, improved understanding of mercury transport and fate is a high priority particularly as international developments in mercury control through United Nations processes accelerate. It is widely recognized that southern hemisphere measurements are inadequate so these measurements are particularly valuable as only a limited number of sites are monitoring GEM in the SH. However, when compared to other articles published regarding GEM measurements in the SH, this manuscript does not offer anything new. (See Brunke et.al, from Cape Point, Slemr et al. Agnot et al. for Amsterdam Island). The main downfall of the paper is that only GEM and meteorological parameters were measured during the 4-year period thus making the conclusions from this paper very straight forward. No new findings or revelations. As ACP is a highly rated scientific journal (Impact Factor of 5.6) I would suggest that the authors revised the manuscript to the level of papers published in this journal before submitting for a second round of review. What the authors could do is discuss the comparison of their GEM measurements with other GMOS sites in the SH in a bit more detail. The authors state in the abstract that that GEM concentrations ranged between 0.23 – 1.43 ng/m3. Did they investigate why this low GEM concentration was observed? Was this a type of depletion event ect.